# Clinical, Radiographic, and Biomechanical Evaluation of the Upper Extremity in Patients with Osteogenesis Imperfecta

**DOI:** 10.3390/jcm13175174

**Published:** 2024-08-31

**Authors:** Katharina Oder, Fabian Unglaube, Sebastian Farr, Andreas Kranzl, Alexandra Stauffer, Rudolf Ganger, Adalbert Raimann, Gabriel T. Mindler

**Affiliations:** 1Department of Pediatric Orthopaedics, Orthopaedic Hospital Speising, Speisinger Strasse 109, 1130 Vienna, Austriaalexandra.stauffer@oss.at (A.S.); r.ganger@aon.at (R.G.); gabriel.mindler@oss.at (G.T.M.); 2Laboratory for Gait and Movement Analysis, Orthopaedic Hospital Speising, Speisinger Strasse 109, 1130 Vienna, Austria; fabian.unglaube@oss.at (F.U.); andreas.kranzl@oss.at (A.K.); 3Vienna Bone and Growth Center, Vienna, Austria, Währinger Gürtel 18–20, 1090 Vienna, Austria; adalbert.raimann@meduniwien.ac.at; 4Department of Pediatrics and Adolescent Medicine, Division of Pediatric Pulmonology, Allergology and Endocrinology, Medical University of Vienna, Währinger Gürtel 18–20, 1090 Vienna, Austria

**Keywords:** osteogenesis imperfecta, upper extremity, motion analysis, deformity

## Abstract

**Introduction:** Osteogenesis imperfecta (OI) is a hereditary disorder primarily caused by mutations in type I collagen genes, resulting in bone fragility, deformities, and functional limitations. Studies on upper extremity deformities and associated functional impairments in OI are limited. This cross-sectional study aimed to evaluate upper extremity deformities and functional outcomes in OI. **Methods:** We included patients regardless of their OI subtypes with a minimum age of 7 years. Radiographic analysis of radial head dislocation, ossification of the interosseous membrane, and/or radioulnar synostosis of the forearm were performed, and deformity was categorized as mild, moderate, or severe. Clinical evaluation was performed using the Quick Disabilities of Arm, Shoulder, and Hand (qDASH) questionnaire and shoulder-elbow-wrist range of motion (ROM). Three-dimensional motion analysis of the upper limb was conducted using the Southampton Hand Assessment Procedure (SHAP). The SHAP quantifies execution time through the Linear Index of Function (LIF) and assesses the underlying joint kinematics using the Arm Profile Score (APS). Additionally, the maximum active Range of Motion (aRoM) was measured. **Results:** Fourteen patients aged 8 to 73 were included. Radiographic findings revealed diverse deformities, including radial head dislocation, interosseous membrane ossification, and radioulnar synostosis. Six patients had mild, six moderate, and two severe deformities of the upper extremity. Severe deformities and radial head dislocation correlated with compromised ROM and worse qDASH scores. The qDASH score ranged from 0 to 37.5 (mean 11.7). APS was increased, and LIF was reduced in OI-affected persons compared with non-affected peers. APS and LIF also varied depending on the severity of bony deformities. aRoM was remarkably reduced for pro-supination. **Conclusion:** Patients with OI showed variable functional impairment from almost none to severe during daily life activities, mainly depending on the magnitude of deformity in the upper extremity. Larger multicenter studies are needed to confirm the results of this heterogeneous cohort. **Level of evidence:** Retrospective clinical study; Level IV.

## 1. Introduction

Osteogenesis imperfecta (OI) describes a group of inherited disorders, most often due to mutations in type I collagen genes (COL1A1, COL1A2) by an autosomal dominant inheritance [1]. It is a rare disorder (one in 15–20,000 births) [2] and is characterized by low bone mass, bone fragility, multiple fractures and deformities of the extremities and spine [3]. In addition, extraskeletal manifestations such as hyperlaxity of ligaments and skin, hearing impairment, blue sclerae and dentinogenesis imperfecta may also be associated [1]. In 1979, Sillence et al. defined four subtypes (OI type I–IV) based on clinical and genetic features [4]. This classification system was expanded by other mostly autosomal-recessive types [5]

Lower limb deformities are a primary cause of reduced mobility in Osteogenesis imperfecta, and functional impairment of the lower limbs has been described in prior studies [6,7,8]. Although quite sparse, recent studies for upper extremity deformities in OI describe impaired function, especially in patients with severe deformities [9,10,11]. However, none of the previous studies regarding upper extremity pathology in OI cases included a 3D motion analysis. This technology is a valuable tool for quantifying functional movement in humans. In clinical orthopedics, it has been well-established for over 30 years, particularly for analyzing lower extremity movements, such as gait analysis [12]. In recent years, there has been an increasing application of 3D motion analysis in evaluating upper extremities, particularly in conditions such as cerebral palsy [13]. In our institution, we frequently use this analysis to evaluate the course of the disease and outcomes of operated and non-operated patients for several other pathologies. Our aim was thus to evaluate the severity of deformity in the upper extremity, both clinically and radiographically, in a consecutive patient cohort and to study the impact of functional impairment on daily life activities.

## 2. Materials and Methods

### 2.1. Study Design and Patients

This retrospective clinical study was approved by our institutional ethics board (EK 20/2021). Written patient consent was obtained prior to the initiation of the analysis. We included patients who were diagnosed with OI regardless of their subtypes and operated or non-operated status on the upper extremity and with a minimum age of 7 years. The patients were examined once after being invited by mail within a period of 6 months. No upper age limit was considered for this study. Exclusion criteria were pregnancy and recent fractures of the upper extremity (non-consolidated) in the last 24 months.

### 2.2. Radiographic Assessment

For each patient, standardized anteroposterior and lateral radiographs of the humerus and forearms were obtained. We screened the radiographs for typical deformities in OI, such as diaphyseal bowing, radial head dislocation, ossification of the interosseous membrane and/or radioulnar synostosis of the forearm and ulna minus deformity. Since the pathologies of humerus, radius, and ulna may be hard to compare among each other, and with degrees of axial bowing likely being clinically less relevant, we decided to grade the radiographic deformities of all above-mentioned bones arbitrarily into either mild, moderate, or severe. Group 1 comprised those with mild (deformity angle 1–15 degrees), group 2 those with moderate (deformity angle 16–30 degrees), and group 3 those with severe deformities (over 30 degrees of deformity angle), respectively.

### 2.3. Clinical Assessment

The Quick Disabilities of Arm, Shoulder and Hand (qDASH) questionnaire was used to analyze a patient’s ability to complete daily life and the severity of symptoms in the upper extremity [14]. It contains 11 items and a five-point Likert scale. Depending on the functional level, the patient can choose from “no difficulty/no symptoms” to “unable to do/extreme symptoms”. A score of 0 means no, and 100 is the most severe disability. Furthermore, range-of-motion (ROM) assessments for the shoulder, elbow, and wrist joints were conducted once with a manual goniometer by an impartial observer not involved in surgical treatment who is also a fellowship-trained pediatric orthopedic surgeon specializing in the upper limb differences, deformities, and dysplasia and routinely involved in patient care in this field.

### 2.4. 3D Motion Analysis

A marker set consisting of sixty-three retro-reflective markers (32 anatomical and 31 technical) was applied to the patient’s upper body as described by Van Andel et al. [15]. Due to expected soft tissue artifacts, we adopted the marker set also described by Van Andel et al. [15], excluded the scapula, and defined a clavicle segment. The marker trajectories were recorded using a 3D motion capture system (17 cameras, 150 Hz, Vicon, Oxford, UK) to quantify upper extremity joint kinematics. Data preprocessing was conducted with Vicon Nexus Version 2.10.2 (Vicon, Oxford, UK), including trajectory reconstruction, labeling, gap filling, and filtering (as described by Woltring et al.) [16]. The marker trajectories were further processed using the open-source custom-made software “Upper Limb Evaluation in Movement Analysis” [17,18] within MATLAB 2022a (The MathWorks, Massachusetts, USA). The local coordinate systems of the segments and joint centers were defined in accordance with the guidelines provided by the International Society of Biomechanics (ISB) as mentioned in Wu et al. [19].

To quantify the function and kinematics of individuals with osteogenesis imperfecta (OI), participants underwent the following:Southampton Hand Assessment Procedure (SHAP) [20];Measurement of maximum active range of motion (aRoM) for the wrist, elbow, forearm, and shoulder in all planes.

The SHAP quantifies hand function through 26 tasks (see Table 1), replicating activities of daily living [20]. We recorded one valid trial with the dominant side for all 26 tasks. SHAP outcomes include the Linear Index of Function (LIF) and the Arm Profile Score (APS). LIF quantifies the execution time in relation to healthy subjects (HSs) [21]. The LIF serves as a transparent alternative to the original score used, known as the Index of Function (IoF), as described in Light et al. [20]. LIF score ranges from 0 to 100. A lower value indicates a slower execution time [21]. Based on the LIF, the SHAP quantifies hand function; however, it provides limited information about underlying joint kinematics. The APS quantifies joint kinematic deviation compared with healthy subjects (HSs), as described by Jaspers et al. [18]. A higher APS in an affected individual, when compared to the non-affected group, indicates greater deviations in joint angle amplitudes. LIF and APS were calculated for each of the 26 tasks and as a mean value per participant.

The maximum aRoM for the assessed joints and degrees of freedom (Table 1) was determined by conducting five valid trials for both sides. aRoM was calculated as the range between the minimum and maximum amplitudes of the mean curve derived from these trials. The comparative dataset (number: 20; age: 29.3 ± 9 y; body height: 176 ± 98 cm; body weight: 73.4 ± 10.3 kg) for the non-affected persons concerning SHAP and aRoM was previously recorded in our laboratory for clinical and research purposes [22].

### 2.5. Statistical Analysis

Since the patient cohort was rather small and heterogenous in age and body size, a purely descriptive analysis using means and ranges was performed with MATLAB 2022a (The MathWorks, Natick, MA, USA). To assess function and kinematics in relation to the severity of the underlying bony deformities, we categorized LIF, APS, and aRoM values for the limb under assessment based on the severity of the bony deformities determined from radiographs.

## 3. Results

A total of 14 people (six male and eight female) agreed to participate and were included in the study. The patient’s ages ranged from 8 to 73 (mean age, 35 years). Using the modification of the Sillence classification by Glorieux et al. [23], eight patients were classified as type 1, one as type 4, and two as type 5. Two patients clinically appeared as type I but genetically remained unclassified. Another patient had Bruck Syndrome (genetically verified but atypical Bruck syndrome without joint contractures).

### 3.1. Radiographic Results

With regard to the radiographic findings, we observed the following results listed in Table 2. Six patients had mild, six a moderate, and two a severe deformity of the upper extremity. The deformities in the upper extremity are demonstrated in Table 3. Two patients with severe deformity were both classified as OI type V and also showed anterior/anterolateral radial head dislocation and interosseous membrane ossification/radioulnar synostosis.

### 3.2. Clinical Results

We subclassified the scores according to the radiologic deformity group of the upper extremity. The shoulder, elbow and wrist ROM values are highlighted in Table 4. The qDASH Score ranged from 0 to 37.5 (mean 11.7). The Sports/Performing Arts and Work module indicated median values of 3.13 (range, 0–31.2) and 7.03 (range, 0–25), respectively. In Table 5, we subclassified the score according to the patient’s radiologic deformity of the upper extremity. Overall, these results indicated that patients with mild and moderate deformities have better clinical outcomes than patients with severe deformities.

### 3.3. 3D Motion Analysis

All participants completed the required tasks except for one young individual who declined to participate in the SHAP due to non-compliance (OI type I, mild deformity group).

In the SHAP, the OI group exhibited higher mean values in both the overall APS (OI mean: 19°, HS mean: 5.1°) and each SHAP task compared to the HS group. All tasks and the overall APS for the OI group deviated more than 2 std from the HS mean. We detected lower LIF values for both the overall LIF (OI mean: 95, HS mean: 100) and each task. The overall LIF and each task except “carton pouring” fell outside 1 std of the HS mean.

Figure 1 and Figure 2 illustrate the APS and LIF in relation to the underlying bony deformities.

Regarding the measurement of aRoM, the OI group fell within one standard deviation of the HS mean across all joints and degrees of movement, except for pro-supination. Pro-supination in the OI group was lower than in healthy subjects, falling outside 1 std of the HS mean but still within two standard deviations (OI: 85.5° ± 22°, HS: 113.9° ± 16.7° [mean ± 1 std]). In OI, shoulder rotation also exhibited a notable decrease (OI: 106.9° ± 26.7°, HS: 117.3° ± 20.4°) but stayed within 1 std of the HS mean (detailed results: Appendix A). Pro-supination aRoM in relation to the bony deformities is illustrated in Figure 3.

## 4. Discussion

To the best of our knowledge, this is the first report addressing the upper limb function and movement with the help of 3D motion analysis in patients with OI. Our aim was to evaluate the severity of deformity in the upper extremity, both from clinical and radiological perspectives and to study the impact of functional impairment on daily life activities.

Radiographically, within this cohort of patients, deformities were evident in several patients. Notably, four patients exhibited radial head dislocations, with the direction being in two patients anterior/anterior-lateral and in the others posterior. Among our cohort, three patients were diagnosed with type 5 OI, two of whom manifested radial head dislocations, both occurring in an anterior/anterior-lateral direction. This alignment, as highlighted by Fassier et al., is typical of type V OI. Moreover, their study identified a distinct association between radial head dislocation or subluxation in type V and calcification of the interosseous membrane [10]. Confirming this finding, two patients (66%) diagnosed with type V exhibited this association.

Our analysis revealed bone deformities classified as mild in six, moderate in the other six and severe in two patients. Regarding the humerus, predominantly mild deformities were observed. In the radius and ulna, mainly mild and moderate deformities were prevalent. Noteworthy are two patients exhibiting severe deformities, both classified as OI type V, coinciding with radial head dislocation and interosseous membrane calcification or radioulnar synostosis. In two other patients with radial head dislocation, the deformity was graded as mild and moderate. Amako et al. showed that upper limb deformities are common in children, rare among OI type I, frequent and severe in OI type II, and moderate or mild in OI types IV and V [9]. Moreover, Fassier’s work indicated that forearm bone bowing accompanies frequent radial head malalignment [10]. Consequently, severe deformity might not only be associated with OI type 5 in our patients but also with radial head dislocation.

Only sparse literature exists about the functional outcome of upper extremity in patients with OI [9,10]. In contrast, Amako et. al. also evaluated the functional outcome using the Pediatric Evaluation of Disability Inventory (PEDI). Their results indicate a significant negative correlation between PEDI scores and total deformity angles. Notably, in cases of no, mild, and moderate deformities, the self-care scores did not show any differences, whereas a decline was evident in severe deformity. Mobility function was dramatically impaired in both moderate and severe deformities [9]. Fassier et al. explored the correlation of forearm range of motion in pro-/supination and hand grip force and different OI groups alongside deformity parameters. They showed that patients with radial head dislocation had a functional impairment, particularly in reduced range of motion and grip force [10]. Our data align with these observations, indicating that severe deformity and radial head dislocation correlate with limitations in elbow movements—especially in supination and pronation. This finding is echoed in higher DASH scores in patients with severe deformity.

Previous research demonstrated SHAP’s ability to identify functional disabilities. Kyberd et al. showed the utility of the SHAP in various orthopedic cases involving the hand and forearm [24]. In our laboratory, we applied the SHAP to individuals treated with the single-bone-forearm procedure [22]. To quantify the function and kinematics of the upper extremities in individuals affected by OI, we used the LIF derived from the SHAP. For all tasks, we observed lower LIF values in our OI-affected cohort. These findings imply an extended execution time and, thus, reduced hand function within our OI-affected cohort.

Joint kinematic analysis revealed increased APS across all tasks, indicating greater deviations in OI individuals compared with healthy subjects, possibly linked to compensatory movements due to functional impairments. Kinematic deviations may impact pain, overuse injuries, and quality of life in individuals with upper limb deficiency [25].

We observed a remarkable reduction in pro-supination and a minor decrease in shoulder rotation while assessing the aRoM. However, all other joints and degrees of freedom showed no remarkable alterations. In APS and LIF derived from SHAP tasks that involve a high demand on transversal hand motion (e.g., page-turning), we did not detect notable differences between OI and HS groups (Figure 4). This could be explained either with a lower necessary range of motion to complete the task or compensatory movements. Therefore, it appears that OI-affected persons do not exhibit significantly impaired function in specific activities of daily living attributed to a limited active pro-supination.

Regarding bony deformities, our data indicate a non-linear relationship between the severity of the underlying bony deformities and active pro-supination. However, we observed a linear trend indicating reduced function and altered kinematics with increased deformity severity in the SHAP (Figure 3). Those findings partly reflect the aforementioned findings of Amako et al. [9]. In the future, it would certainly be desirable to integrate 3D motion analysis into the clinic in order to objectively assess OI patients’ functional impairments in the upper extremities and to initiate specific conservative and surgical interventions based on that assessment. Especially in rehabilitation, it could be helpful for therapists and doctors to better understand the limitations of patients and to be able to offer more specific therapies.

We acknowledge limitations in our study, including small sample sizes and variability in age and OI types, as well as no information on prior surgical treatment, fractures, age-related changes, and limb dominance. Instead of genetic nomenclature, we used the traditional classification for OI. The use of QuickDASH scores for adults in our study differs from previous pediatric OI research using the PEDI score, limiting direct comparability. The SHAP assessment setup lacks scaling for body size, as highlighted by Vasluian et al., impacting our heterogeneous sample [26]. Besides that, we assessed only the dominant side with the SHAP test, whereas all other investigations were conducted on both sides.

Due to these limitations, we refrained from inferential statistical analysis, hindering generalization. Despite this, our study recruited a decent sample based on known incidences.

## 5. Conclusions

In conclusion, patients with OI showed variable functional impairment from almost none to severe during daily life activities, mainly depending on the magnitude of deformity in the upper extremity. Larger multicenter studies are needed to confirm the results of this heterogeneous cohort.

## Figures and Tables

**Figure 1 jcm-13-05174-f001:**
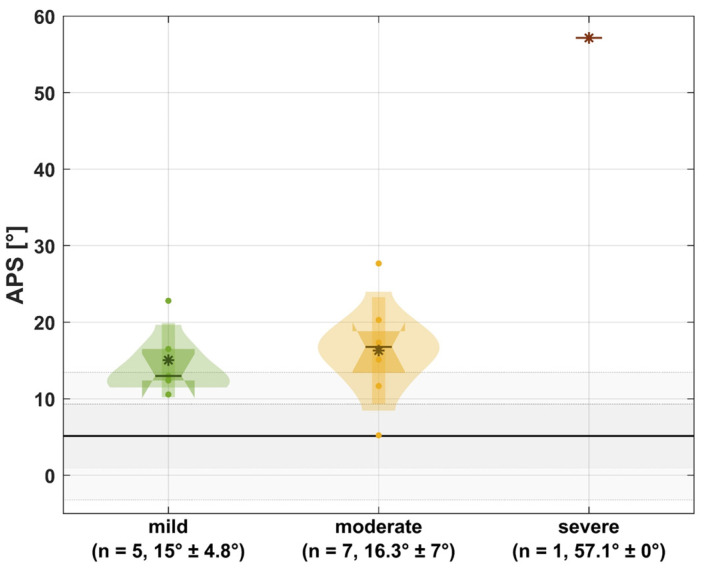
APS in relation to the underlying bony deformities. Gray: mean of non-affected persons with one and two times standard deviation; *: mean; -: median; small box: one times standard deviation; wide box: interquartile range; n: number of extremities; mean ± standard deviation.

**Figure 2 jcm-13-05174-f002:**
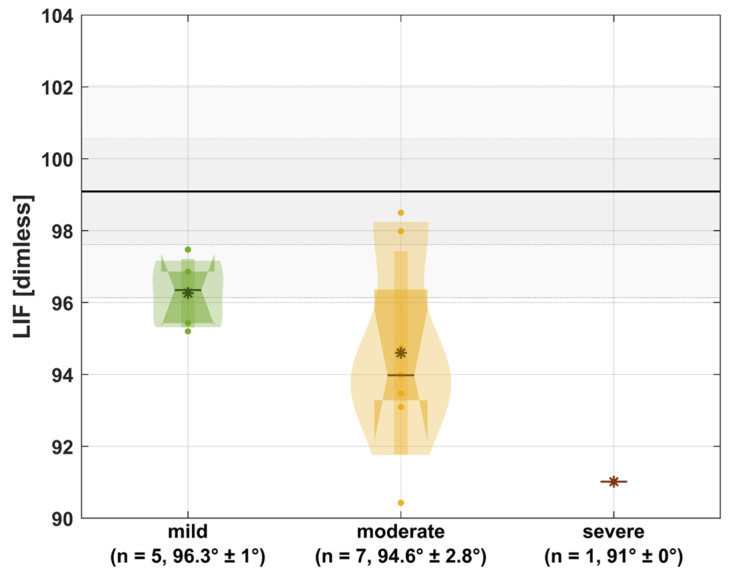
LIF in relation to the underlying bony deformities. Gray: mean of non-affected persons with one and two times standard deviation; *: mean; -: median; small box: one times standard deviation; wide box: interquartile range; n: number of extremities; mean ± standard deviation.

**Figure 3 jcm-13-05174-f003:**
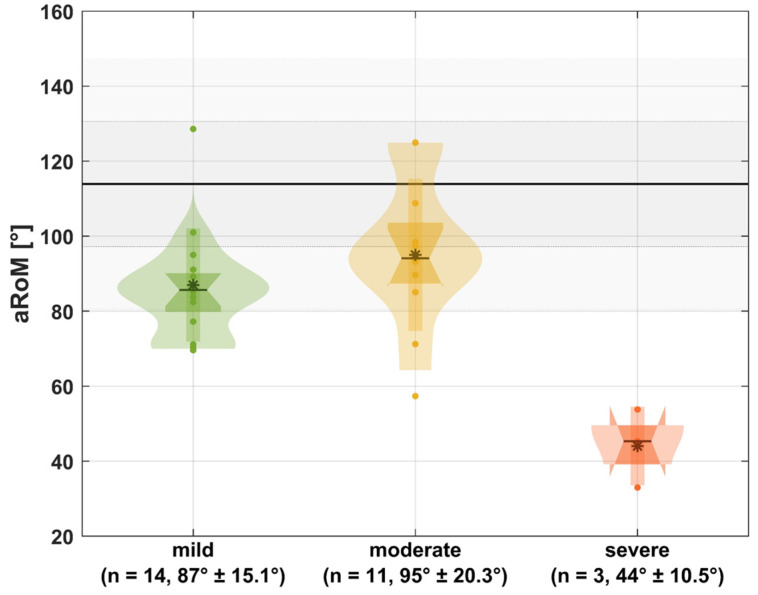
Pro-supination aRoM in relation to the bony deformities. Gray: mean of non-affected persons with one and two times standard deviation; *: mean; -: median; small box: one times standard deviation; wide box: interquartile range; n: number of extremities; mean ± standard deviation.

**Figure 4 jcm-13-05174-f004:**
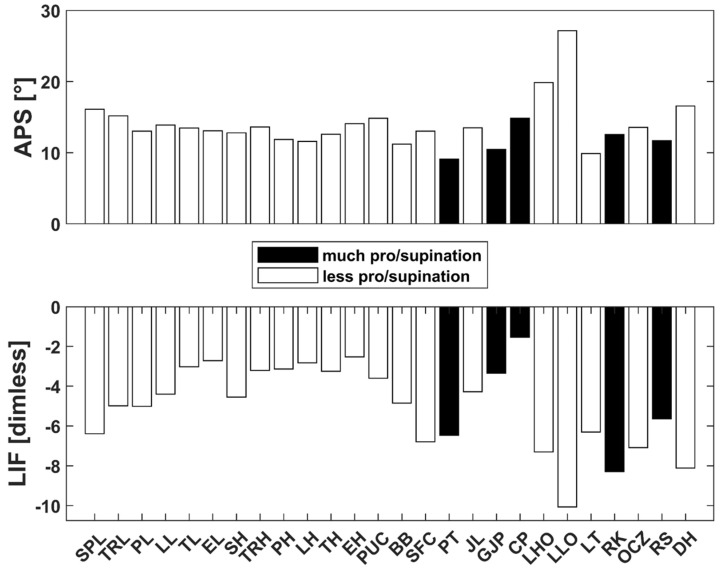
Differences between OI-affected and non-affected persons for all SHAP tasks. The black bars indicate tasks that require more forearm supination/pronation.

**Table 1 jcm-13-05174-t001:** SHAP tasks and examined joints for aROM.

		SHAP				aROM	
		*object tasks (tasks 1–12)*			shoulder sagittal (flexion)	shoulder frontal (ab/adduction)	shoulder transversal (ext./internal rotation)
	spherical light (SPL)	tripod light (TRL)		power light (PL)	elbow sagittal (flexion/extension)		elbow transversal (pro/supination)
	lateral light (LL)	tip light (TL)		extension light (EL)	wrist sagittal (flexion/extension)	wrist frontal (ulnar/radialadduction)	
	spherical heavy (LH)	tripod heavy (TRH)		power heavy (PH)			
	lateral heavy (LH)	tip heavy (TH)		extension heavy (EH)			
		*activities of daily living (tasks 13–26)*					
	pick up coin (PUC)	button board (BB)		simulated food cutting (SFC)			
	page-turning (PT)	jar lid (JL)		glass jug pouring (GJP)			
	carton pouring (CP)	lifting a heavy object (LHO)		lifting a light object (LLO)			
	lifting a tray (LT)	rotate key (RK)		open/close zip (OCZ)			
	rotate a screw (RS)	door handle (DH)					
*Outcomes*	LIF [dimless]		APS [°]		aROM [°]		

SHAP: Southampton Hand Assessment Procedure; aROM: active Range of Motion; LIF: Linear Index of Function; APS: Arm Profile Score.

**Table 2 jcm-13-05174-t002:** Radiographic findings.

Radiological Findings	Number of Patients in Different OI Types				
	I	III	IV	V	Unclassified
Anterior radial head dislocation				2	
Posterior radial head dislocation	1				1
Interosseous membrane ossification				1	1
Radioulnar synostosis				1	
Ulnar variance	2				

**Table 3 jcm-13-05174-t003:** Deformities in upper extremity.

Grade of Deformity	Degrees of Deformity					
	Humerus		Radius		Ulna	
	Right	Left	Right	Left	Right	Left
Mild	13	14	7	7	8	9
Moderate	1	0	6	7	5	3
Severe	0	0	1	0	1	2

**Table 4 jcm-13-05174-t004:** The shoulder, elbow, and wrist ROM values.

Motion	Mild Deformity	Moderate Deformity	Severe Deformity	Patient without Radial Head Dislocation	Patient with Radial Head Dislocation
Shoulder abduction	176 (170–180)	178 (160–180)	177 (170–180)	178 (170–180)	171 (160–180)
Shoulder adduction	33 (20–40)	34 (20–40)	37 (30–40)	35 (20–40)	29 (20–40)
Shoulder retroversion	40 (40–40)	39 (30–40)	35 (30–40)	40 (30–40)	36 (30–40)
Shoulder anteversion	169 (150–180)	169 (140–180)	168 (160–180)	170 (150–180)	161 (140–170)
Shoulder external Rotation	76 (50–95)	73 (50–95)	53 (45–65)	73 (45–95)	68 (50–85)
Shoulder internal Rotation	69 (50–80)	75 (70–80)	70 (70–70)	71 (50–80)	75 (70–80)
Elbow extension	0 (−45–10)	12 (0–25)	3 (0–10)	7 (−20–25)	–6 (−45–10)
Elbow flexion	134 (130–135)	133 (110–135)	110 (100–130)	133 (100–135)	120 (100–135)
Forearm external rotation	77 (80–90)	80 (0–90)	25 (10–45)	75 (10–90)	56 (0–90)
Forearm internal rotation	88 (50–100)	89 (45–100)	48 (10–90)	88 (45–100)	61 (10–100)
Hand dorsiflexion	68 (35–80)	74 (50–100)	73 (65–85)	71 (35–100)	69 (50–100)
Hand palmarflexion	79 (70–90)	84 (65–100)	57 (45–75)	80 (50–00)	73 (45–100)
Hand radialduction	31 (20–45)	21 (0–35)	22 (10–35)	25 (0–45)	30 (20–35)
Hand ulnarduction	38 (25–45)	40 (30–45)	42 (40–45)	39 (25–45)	41 (40–45)

ROM: Range of Motion.

**Table 5 jcm-13-05174-t005:** qDASH subclassified according to patients’ radiologic deformity.

Grade of Deformity		qDASH	Work Module	Sports/Arts Module
Mild	Mean	8.3	14.6	0
	Min	0	0	0
	Max	34	25	0
Moderate	Mean	10.4	0	0
	Min	0	0	0
	Max	37.5	0	0
Severe	Mean	26.1	6.3	12.5
	Min	25	0	0
	Max	27.2	12.5	25

qDASH: Quick Disabilities of Arm, Shoulder, and Hand (qDASH) questionnaire, Min: Minimum value; Max: Maximum value.

## Data Availability

The original contributions presented in the study are included in the article/Appendix A, further inquiries can be directed to the corresponding author/s.

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
