# Peer review of "Clinical, Radiographic, and Biomechanical Evaluation of the Upper Extremity in Patients with Osteogenesis Imperfecta"

_jcm, 2024, doi:10.3390/jcm13175174_

Round 1

Reviewer 1 Report

Comments and Suggestions for Authors

The manuscript “Clinical and biomechanical evaluation of the upper extremity 2 in patients with osteogenesis imperfecta”, by Oder at al, aimed to evaluate upper extremity deformities and functional outcomes in patients with osteogenesis imperfecta (OI). Results showed diverse deformities, with severe deformities and radial head dislocation correlating with compromised range of motion (ROM) and worse qDASH scores. Authors revealed that functional impairment varied widely among patients, depending on the severity of the deformity, highlighting the need for larger multicenter studies to confirm these findings. However, I have a number of concerns.

1.       The major comment is the lack of statistical analysis and statistical significance assessment. Without comparing statistical significance and P-value, all comparisons between groups have no scientific sense.

2.       The sample size is extremely small and is further subdivided by age and gender, which also does not allow us to adequately evaluate the results obtained. the sample is extremely small and is further subdivided by age and gender, which also does not allow us to adequately evaluate the results obtained.

3.       Moreover, the manuscript requires detailed justification that age-related changes do not affect the clinical indicators that were determined within the framework of radiological and clinical assessments.

4.       Why the diagnosis criteria was based on Sillence classification, if genetic analysis was performed? Genetic nomenclature is more accurate and when determining the types of OI, it is better to use it, as sometimes they are not equivalent.

5.       In addition, it is better to present the results of genetic testing (identified mutations), which will significantly strengthen the manuscript and allow us to compare approaches to determining the type of OI.

6.       No information on HS group is provided in manuscript – who are them? Do they correspond to the patient’s group in age, sex, etc.?

Minor comments:

Line 34, the presented prevalence of OI types correspond to Swedish population, this should be mentioned in the text, or better provide a link to an international population study

Comments on the Quality of English Language

Lines 36-37, re-check and rephrase, «Besides that, extraskeletal manifestations can be associated such as hyperlaxity of ligaments and skin, hearing impairment, blue sclerae and dentinogenesis imperfecta».

Author Response

The research aims to evaluate upper extremity deformities and functional outcomes in patients with osteogenesis imperfecta. The affiliations of the authors should be noted. The tile should be more specific as radiographic assessment was made in this study; the words should start with uppercase, except the linking words. The abstract is structured; the keywords should checked in accordance with MeSH.

Response: Thank you for your thorough review and suggestions. We added the affiliations and changed the title. Further we checked the keywords.

The introduction should be extended and include information about 3D motion analysis and evaluation means of the disease.  

Response: Thank you for your suggestion, we added some more information about 3 D motion analysis.

In the methodology section, the stages of the research are presented in an organized manner. Subsection 2.1 should be named “ Study Design and Patients”; the authors should present the type of study and timeframe also here. Line 87/88 – please present the manufacturer and country; line 91 – please present the state when using manufacturers from the USA. What software was used for the descriptive statistics? The results are clear; there is misplacement of the figures and tables.

Response: Thank you for your recommendations. We changed Subsection 2.1 and added the missing information. Also we changed the acquired information in line 87/88 and 91. We also added timeframe. The figure arrangement has been corrected but might still be applicable for further editorial edits and re-arrangements.

In the discussion section Table 4 should be presented at the results. A discussion on rehabilitation ostentations as future research perspective could enhance the value of this section, in relation to the scientific literature (for e.g. https://doi.org/10.1016/j.promfg.2018.03.122 ). Limitations of the study are provided at the end of the section.

Response: Thank you for your advice: We think instead of Table 4, Figure 4 was meant. We present this Figure in the discussion because this is an alternative presentation and combination of the previously presented results that emerged in the discussion of the descriptive statistics clearly defined in the methods section. The aim is to support the discussion of the clinical relevance of the results.

In the future, it would certainly be desirable to integrate 3D motion analysis into the clinic in order to objectively assess OI patient's functional impairments and to initiate specific conservative and surgical interventions based on that assessment. Especially in rehabilitation, it could be helpful for therapists and doctors to better understand the limitations of patients and to be able to offer more specific therapies. Thank u for that important advice.

Are the conclusions supported by the results?

Response: we added the conclusions in the end of the discussion, and yes, the conclusions are definitely supported by the results.

A clear and concise conclusions section is missing.

Response: we added the conclusions in the end of the discussion

The references are adequate but should be extended as suggested above as 24 is not enough for a research article.

Response: Thank you, we added some references.

Editing recommandation – the references should be noted with ”[ ]” rather than in superscript; the reference number should be noted next to the author name (for e.g. line 38 ”Sillence et al. [4]”)

Response: Thank you, we changed it accordingly.

Reviewer 2 Report

Comments and Suggestions for Authors

1. Please adhere to the EQUATOR guidelines for observational studies and report the items listed in the guidelines.

2. Please submit the filled-in EQUATOR guideline checklist mentioning the items listed in the checklist corresponding to the ones listed in the manuscript.

3. Please provide the appropriate citation to the statement defining the severity of deformities mentioned in the methods section.

4. Mention the details of sample size estimation in the statistical analysis section.

5. State the names of statistical tests used for comparison of outcomes between the groups in this study.

Author Response

  1. Please adhere to the EQUATOR guidelines for observational studies and report the items listed in the guidelines.

Response: Thank you for these guidelines, we checked them accordingly

  1. Please submit the filled-in EQUATOR guideline checklist mentioning the items listed in the checklist corresponding to the ones listed in the manuscript.

Response: Thank you for advice. Here we submit de filled- in guideline checklist.

  1. Please provide the appropriate citation to the statement defining the severity of deformities mentioned in the methods section.

Response: Thank you for your comment. We defined this classification arbitrary based on our extensive clinical experience in order to better compare our results.

  1. Mention the details of sample size estimation in the statistical analysis section.

Response: Thank you for your advice. Since we performed a descriptive exploratory analysis without in-depth statistical analysis, we do not necessarily need any sample size estimation for this work.

  1. State the names of statistical tests used for comparison of outcomes between the groups in this study.

Response: Thank you for your advice. Our cohort was small and very heterogenous. Thus we performed a descriptive analysis without statistical analysis.

Reviewer 3 Report

Comments and Suggestions for Authors

The research aims to evaluate upper extremity deformities and functional outcomes in patients with osteogenesis imperfecta. The affiliations of the authors should be noted. The tile should be more specific as radiographic assessment was made in this study; the words should start with uppercase, except the linking words. The abstract is structured; the keywords should checked in accordance with MeSH.

The introduction should be extended and include information about 3D motion analysis and evaluation means of the disease.

In the methodology section, the stages of the research are presented in an organized manner. Subsection 2.1 should be named “ Study Design and Patients”; the authors should present the type of study and timeframe also here. Line 87/88 – please present the manufacturer and country; line 91 – please present the state when using manufacturers from the USA. What software was used for the descriptive statistics? The results are clear; there is misplacement of the figures and tables.

In the discussion section Table 4 should be presented at the results. A discussion on rehabilitation ostentations as future research perspective could enhance the value of this section, in relation to the scientific literature (for e.g. https://doi.org/10.1016/j.promfg.2018.03.122 ). Limitations of the study are provided at the end of the section.

A clear and concise conclusions section is missing.

The references are adequate but should be extended as suggested above as 24 is not enough for a research article.

Editing recommandation – the references should be noted with ”[ ]” rather than in superscript; the reference number should be noted next to the author name (for e.g. line 38 ”Sillence et al. [4]”)

Author Response

  1. The major comment is the lack of statistical analysis and statistical significance assessment. Without comparing statistical significance and P-value, all comparisons between groups have no scientific sense.

Response: Thank you for this comment. Since our cohort was so small, we decided to only describe the results. However, we believe that we have analyzed and presented our data in a very clear and clean descriptive manner.

  1. The sample size is extremely small and is further subdivided by age and gender, which also does not allow us to adequately evaluate the results obtained.

Response: Thank you for this comment. Unfortunately, we must admit that our cohort is rather small and heterogeneous which is not surprising based on the rarity of this disease. However, we believe that we have analyzed and presented the data of our tertiary referreal center in the best way possible to provide further knowledge on this rare but important disorder.

  1. Moreover, the manuscript requires detailed justification that age-related changes do not affect the clinical indicators that were determined within the framework of radiological and clinical assessments.

Response: Thank u for this interesting point. We did not focus on age-related changes. When re-examining our data, we cannot see that age plays a role in the clinical and radiological data. We have both younger patients with OI and older patients who perform very well in the clinical tests.

  1. Why the diagnosis criteria was based on Sillence classification, if genetic analysis was performed? Genetic nomenclature is more accurate and when determining the types of OI, it is better to use it, as sometimes they are not equivalent.

Response: Yes, we agree with you that a genetic nomenclature would be better. When we planned our study in 2021, we decided to use the traditional classification, as is done in other classical OI papers. To use a genetic nomenclature was discussed for the first time at this year's International Conference on Children´s Bone Health (ICCBH). We have been applying it since then. We added this point in the limitations section.

  1. In addition, it is better to present the results of genetic testing (identified mutations), which will significantly strengthen the manuscript and allow us to compare approaches to determining the type of OI.

Response: We decided to use the Sillence classification since it has been the standard so far. This allows for comparison with older data as well. However, as we are still in the process of transition, we are currently unable to contribute to the determination of the types.

  1. No information on HS group is provided in manuscript – who are them? Do they correspond to the patient’s group in age, sex, etc.?

Response: Thank u for your comment. We already shared this information in line 171: The comparative dataset (number: 20; age: 29.3±9 y; body height: 176±98 cm; body weight: 73.4±10.3 kg).

Minor comments:

Line 34, the presented prevalence of OI types correspond to Swedish population, this should be mentioned in the text, or better provide a link to an international population study

Response: Thank you for this minor comment. We changed it and used another scientific source.

Lines 36-37, re-check and rephrase, «Besides that, extraskeletal manifestations can be associated such as hyperlaxity of ligaments and skin, hearing impairment, blue sclerae and dentinogenesis imperfecta».

Response: We changed it to” In addition, extraskeletal manifestations such as hyperlaxity of ligaments and skin, hearing impairment, blue sclerae and dentinogenesis imperfecta may also be associated” and hope this suits better.

Round 2

Reviewer 1 Report

Comments and Suggestions for Authors

In conclusion, it is incorrect to assert something if it is not confirmed and there has not even been an attempt to confirm the associations with statistical data.

if you claim that a number of diagnoses have been genetically verified, why not present this data? it is very important and should be taken into account first

Author Response

We thank the author again for the comment. While genetic testing has been performed at a pediatric institution for some of these cases, it has not been uniformly done because of a lack of consequences for many children and due to parental preferences. Therefore, this data has not been presented. It moreover would not change any meaning or conclusions of this article.

Reviewer 2 Report

Comments and Suggestions for Authors

Thank you for the revision.

Author Response

Thank you for your remarks.

Reviewer 3 Report

Comments and Suggestions for Authors

The paper has been improved although the authors have failed to provide a point by point response to the initial comments. The misplacement of figures still remains. When discussing future research directions and 3D motion analysis please refer to the scientific literature. The final conclusion should be a different section and it should better relate to the findings of the study.

Author Response

We thank the reviewer again for his/her comments. However, we disagree since we have provided a clear point by point response in the previous revision.

The figures have now alle been re-arranged.

Unfortunately, no pertinent literature has been found that can be cited in regards to this topic and issue/future directions.

The conclusion has been put into a dedicated new section as advised. The conclusions are to the best of our knowledge and scientific expertise very sharp and exclusively related to our findings.